# Conception and Experimental Validation of a Standalone Photovoltaic System Using the SUPC5 Multilevel Inverter

Hind El Ouardi [1,*], Ayoub El Gadari [1], Youssef Ounejjar [1,*] and Kamal Al-Haddad [2]

1 Electrical Engineering Department, L'école Supérieure de Technologie (EST), University Moulay Ismail, Meknes 50000, Morocco
2 Ecole de Technologie Superieure (ETS), Montreal, QC 11290, Canada
* Correspondence: hind.elouardi@edu.umi.ac.ma (H.E.O.); ounejjar@gmail.com (Y.O.)

**Abstract:** In this work, an advanced pulse width modulation (PWM) technique was developed to provide the auto-balancing of the capacitors voltages of the five-level split-packed U-Cells (SPUC5) single-phase inverter, and then, the latter was applied to a photovoltaic (PV) system in standalone mode to evaluate its performance in this kind of application. The SPUC5 inverter makes use of only five switches (four active bidirectional switches and one four quadrant switch), one DC source and two capacitors to generate five levels of output voltage and a current with a quasi-sinusoidal waveform which reduces the total harmonic distortion (THD) without the need to add filters or sensors, and also reduces its cost compared to the other multilevel inverters. In the proposed system; the incremental conductance (INC) algorithm is combined with a DC/DC boost converter to reach the maximum power (MP) of the PV array by tracking the MP point (MPP). The offered concept has been constructed and then simulated in the MATLAB/Simulink environment to evaluate its efficiency. According to the results, the self-balancing of the capacitors voltages has been achieved. A comparative study was performed with the traditional PWM technique. The proposed PV system has been validated by experimental results.

**Keywords:** SPUC5 inverter; MPPT technique; incremental conductance; PV system; PWM controller; total harmonic distortion

## 1. Introduction

Renewable energies are characterized by the fact that they are collected from cleaner sources. Moreover, they are inexhaustible and noiseless, unlike fossil energies that pollute the atmosphere and whose quantity on earth is limited. All these reasons give rise to several renewable energy projects in general and to photovoltaic solar energy in particular.

The solar energy field is the most targeted by researchers in alternative energy. There is about $1.2 \times 10^5$ TW of solar irradiance that covers the global surface of the planet [1]. Photovoltaic cells make use of the photo-electric effect to deliver a direct current by collecting solar radiation. This effect enables the cells to directly convert the light energy of photons into electrical energy thanks to a semiconductor material that carries electrical charges. Therefore, irradiation and temperature are two factors that influence the produced power, which requires the use of the MPP tracking (MPPT) controller to extract the maximum power even under changing input conditions [2]. Various algorithms and methods have been developed to reach the MPP of the PV panels.

In [3,4], the perturb and observe (P and O) method is used to reach the MPP, it involves comparing the actual value of the PV array output power with the previous value while perturbing the PV array voltage. While the incremental conductance (INC) is utilized in [5,6], which uses the incremental and the instantaneous conductance of the PV array, the MPP is then reached when the slope of the PV array is null. The hill climbing (HC) technique consists of imposing a perturbation in the cyclic ratio of the boost converter to

find the MPP [7]. In this study, the INC algorithm is employed to optimize the power of the PV array.

The inverter is also one of the important elements in a PV system; it transforms DC voltage into AC voltage to supply the load. There are two types of inverters: two-level inverters [8] and multilevel inverters [9,10]. The latter type produces low harmonic values and achieves high performance compared to the first type. For these reasons, various multilevel inverter topologies have been developed by researchers, such as cascaded H-bridge (CHB) [11], neutral point clamped (NPC) [12,13] and flying capacitor (FC) [14,15]. However, these traditional topologies have many drawbacks. For example, the NPC inverter needs a diversity of diodes. Hence, in these past years, new topologies have been implemented such as packed U-Cells (PUC) [16,17] and SPUC [18] which comes as a modification of the PUC inverter, which needs a limited count of switches and guarantees high performance in both PV systems [19,20].

Several techniques have been developed to control the DC/AC converters. In [21] a control based on the proportional-integral passivity-based controller (PI-PBC) is used to control the inverter. In [22], a technique has been developed to be used with all electrical devices that use the traditional PWM controllers to increase the efficiency and accuracy of the latter.

In [18], the SPUC9 inverter was controlled by the traditional PWM, it uses more redundant states and reduces the number of levels to achieve the self-balancing of capacitors voltages. It allows, in this case, nine levels of operation with three redundant states. However, this method allows THD rite to be more than 4%. In this paper, an advanced PWM control technique has been proposed and applied to the SPUC5 inverter to properly regulate and attain the self-balancing of the capacitors' voltages and an output current with a quasi-sinusoidal waveform, which reduces the THD in comparison with the previous PWM technique.

This article proceeds in the following sections. First, the SPUC5 inverter associated with the proposed control technique is presented in Section 2. Afterward, the proposed PV system is described in Section 3. Subsequently, the results of the simulation are analyzed in Section 4. Finally, the experimental validation is carried out in Section 5.

## 2. The Proposed Control Technique of the SPUC5 Inverter

Among the newest developed multilevel inverters, we find the SPUC inverter. It is highly efficient and very competitive. Indeed, it outputs five voltage levels with only a single DC supply and two capacitors, as illustrated in Figure 1. The self-balancing of the capacitors' voltages is ensured by the proposed control technique.

Table 1 shows the switching states of the SPUC5 inverter, where $VC1 = VC2 = E/2$ (VC1 and VC2 are the voltage of capacitor 1 and capacitor 2, respectively). In order to generate gate pulses, one has to design a specific modulation technique. Consequently, in this paper, the authors propose the following procedure in which a sign function of the reference voltage is used to determine $S_1$.

$$\text{Sign}\,(v_r) = \begin{cases} 1, v_r \geq 0 \\ 0, v_r < 0 \end{cases} \tag{1}$$

$$S_1 = \text{Sign}\,(v_r) \tag{2}$$

Four triangular carriers were designed to provide the remaining gate pulses. Their comparison to the reference voltage gives four signals as follow:

$$Z_1 = \begin{cases} 1, C_{r1} \geq v_r \\ 0, C_{r1} < v_r \end{cases} \tag{3}$$

$$Z_2 = \begin{cases} 1, C_{r2} \geq v_r \\ 0, C_{r2} < v_r \end{cases} \tag{4}$$

$$Z_3 = \begin{cases} 1, C_{r3} \geq v_r \\ 0, C_{r3} < v_r \end{cases} \tag{5}$$

$$Z_4 = \begin{cases} 1, C_{r4} \geq v_r \\ 0, C_{r4} < v_r \end{cases} \tag{6}$$

where $v_r$ presents the reference volage, $Z_1, Z_2, Z_3$ and $Z_4$ are the comparison results between the reference and carried triangular 1, 2, 3 and 4, respectively; $C_{r1}, C_{r2}, C_{r3}$ and $C_{r4}$ present the carried triangular 1, 2, 3 and 4, respectively.

A staircase signal which the image of the voltage load can then be generated by summing these equations as follows:

$$SS = Z_1 + Z_2 + Z_3 + Z_4 \tag{7}$$

Figure 2 illustrates these signals. The remaining gate pulses can then be expressed as follows.

$$S_2 = \overline{Z_4} \tag{8}$$

$$S_3 = 1 - \text{Sign}(v_r) \tag{9}$$

$$S_4 = Z_1 + (Z_2 \oplus Z_3) \tag{10}$$

$$S_5 = (Z_1 \oplus Z_2) + (Z_3 \oplus Z_4) \tag{11}$$

Finally, Figure 2 describes the proposed control technique. It permits the self-balancing of the capacitors' voltages in open loop operation with no need for a sensor.

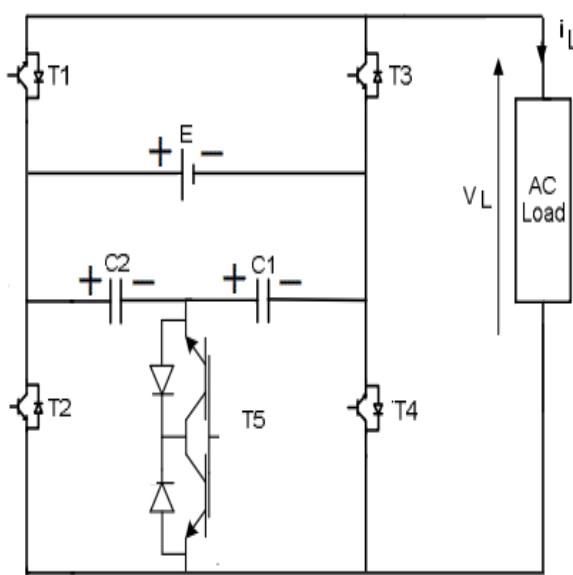

**Figure 1.** The proposed SPUC5 inverter.

**Table 1.** The switching states of the SPUC5 inverter.

| DC Voltage Combination | Switches Pulses | | | | |
|---|---|---|---|---|---|
| | S1 | S2 | S3 | S4 | S5 |
| $E = V_{C1} + V_{C2}$ | 1 | 0 | 0 | 1 | 0 |
| $V_{C1}$ | 1 | 0 | 0 | 0 | 1 |
| 0 | 0 | 0 | 1 | 1 | 0 |
| $-V_{C2}$ | 0 | 0 | 1 | 0 | 1 |
| $-E = -V_{C1} - V_{C2}$ | 0 | 1 | 1 | 0 | 0 |

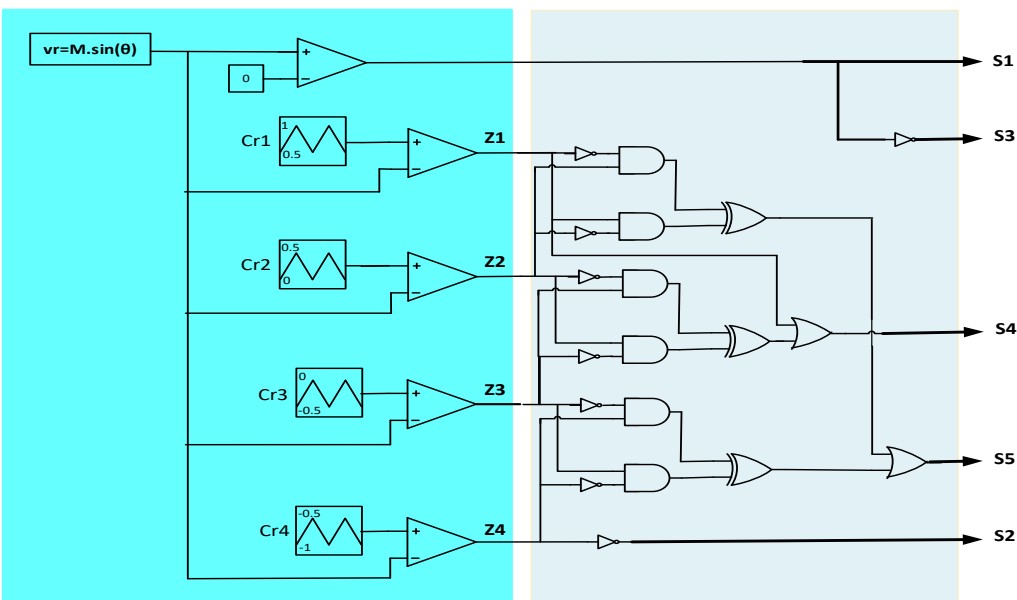

**Figure 2.** The proposed control technique.

## 3. The Suggested Photovoltaic System

The proposed PV system is composed of the PV panels, the DC/DC boost converter with the MPPT controller, and the SPUC5 inverter, which provides an RL load. The resistance and the inductance present the eventual active and passive power consumed by the load, respectively.

### 3.1. The PV Array under Consideration

The PV module is constructed of multiple interconnected cells. The latter is formed by interconnected photodiodes that are capable of converting solar irradiation into a DC current flow, which is called the photovoltaic effect.

A standard PV cell is schematically represented by a source of current linked in parallel with an ideal diode, and two resistors in serial connection, as shown in Figure 3.

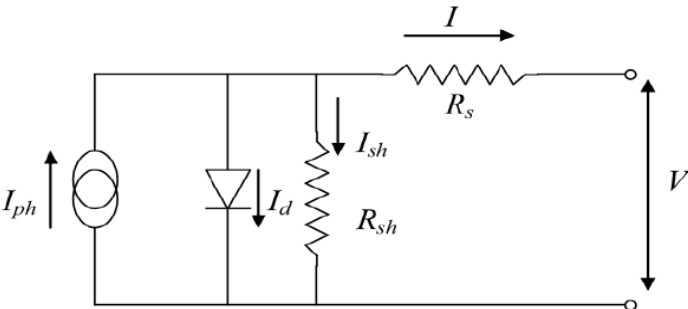

**Figure 3.** A standard PV cell scheme.

Its mathematical model is written in the following equation:

$$I = I_{ph} - I_0 \left[ \exp\left( \frac{V + IR_S}{nV_T} \right) - 1 \right] - \frac{V + IR_S}{R_{Sh}} \tag{12}$$

where $I$ [A] presents the output cell current, $I_{ph}$ [A] is the photocurrent produced by the photons, $I_0$ [A] is the saturation current of the cell during the dark moments, $V$ [V] presents the output voltage of cell, $V_T = kT_c/q$ [V] is the thermal voltage of cell, $n$ is the ideality factor of the diode, $R_s$ [Ω] is the sum of the series resistors, and $R_{sh}$ [Ω] represents the shunt resistance.

Table 2 shows the SunPower SPR-305-WHT panel characteristics. SunPower technology has been preferred because, over its outstanding 35 years of solar expertise, it has been awarded more than 900 patents. Figure 4 shows the P-V and I-V characteristics of the considered PV array and the corresponding MPP values for four various irradiances.

**Table 2.** The PV panel parameters.

| Parameters | Values |
|---|---|
| Maximum current $I_{max}$ at $P_{max}$ (A) | 5.58 |
| Maximum voltage $V_{max}$ at $P_{max}$ (V) | 54.7 |
| Short circuit current $I_{sc}$ (A) | 5.96 |
| Open circuit voltage $V_{oc}$ (V) | 64.2 |
| Series resistance $R_s$ ($\Omega$) | 0.037998 |
| Parallel resistance $R_p$ ($\Omega$) | 993.51 |
| Number of cells | 96 |

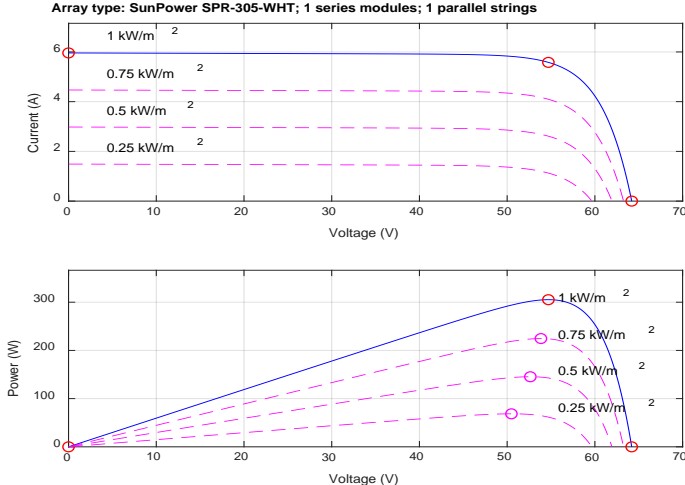

**Figure 4.** I-V and P-V characteristics of one SunPower SPR-305-WHT PV module.

### 3.2. The Utilized MPPT Controller

The DC/DC boost converter offers multiple benefits to PV systems [23]: it regulates the electrical power in the DC chain and magnifies the PV voltage. Duo to varying weather conditions (in terms of irradiation and temperature), P-V and I-V characteristics have unique MP points. The latter should be found and tracked by the designed MPPT technique. The INC technique has been preferred because of its simplicity and accuracy. It is a question of finding the point at which the derivative of power toward voltage is null. Thus, when the MPP is reached, the following condition is verified:

$$\frac{dP}{dV} = 0 \tag{13}$$

To the right of the MPP, the derivative (Equation (13)) is negative, whereas it is positive at the left. This may be written as follows:

$$\begin{cases} \frac{dI}{dV} < -\frac{I}{V} & \text{at right of MPP} \\ \frac{dI}{dV} > -\frac{I}{V} & \text{at left of MPP} \\ \frac{dI}{dV} = -\frac{I}{V} & \text{at MPP} \end{cases} \tag{14}$$

where $\frac{I}{V}$ represents the instantaneous conductance, and $\frac{dI}{dV}$ is the incremental conduction.

The voltage is adapted according to whether the derivative is positive or negative until being null. Therefore, the MPP is achieved and no voltage adjustment is required.

## 4. Simulation Results

The suggested off-grid PV system is verified through the MATLAB/Simulink environment. The PV array is linked to the load through the boost converter and the SPUC5 inverter. The simulation parameters are given in Table 3.

**Table 3.** Simulation parameters.

| Parameters | Values |
| --- | --- |
| Fundamental frequency | 60 Hz |
| Switching frequency | 2000 Hz |
| SPUC5 capacitors | 4000 µF |
| Load resistor | 80 Ω |
| Load inductor | 15 mH |

The proposed system has been verified under an irradiation of 500 W/m$^2$ and a 25 °C of temperature.

Figure 5 shows the progress of the output power of the PV array during the simulation. One can observe that within a short time (less than 0.5 s), the MPP has been attained (around 145 W). The output voltage of the PV array is maintained around the maximum value of 53 V, which coincides with the characteristics of the PV array shown in Figure 4. The evolution of the output voltage of the PV array is shown in Figure 6. It has then been boosted, which produces the DC-link voltage. The evolution of the latter is presented in Figure 7. The DC link voltage is actually the reference for the voltages of the two capacitors, which are properly adjusted and continued around half of the DC-link voltage (75 V).

The load voltage is constituted of five levels as follows: $+V_{dc}$, $+V_{dc}/2$, 0, $-V_{dc}/2$ and $-V_{dc}$ as depicted in Figure 8. Hence, the waveform of the load current is approximately sinusoidal as presented in Figure 9.

These results have been obtained without using any filters or feedback. Moreover, with such waveforms, there is no need for any filters, which reduces the product cost.

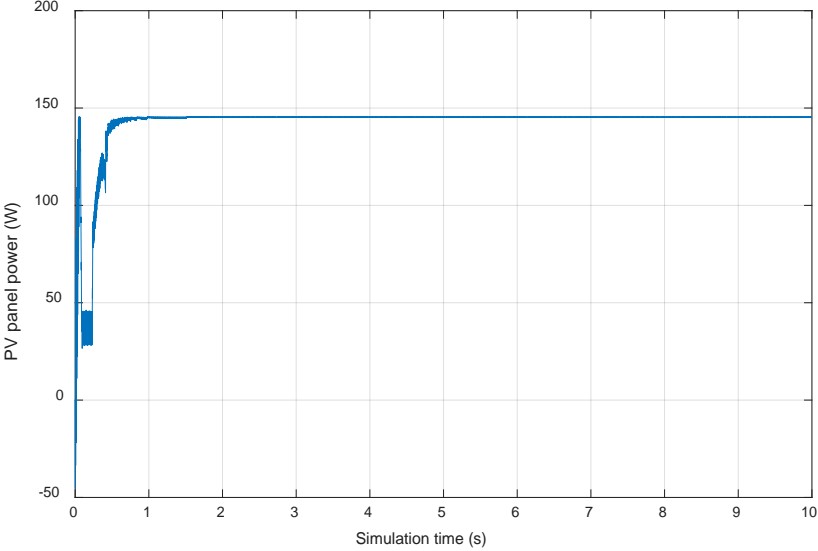

**Figure 5.** PV modules output power at constant irradiation.

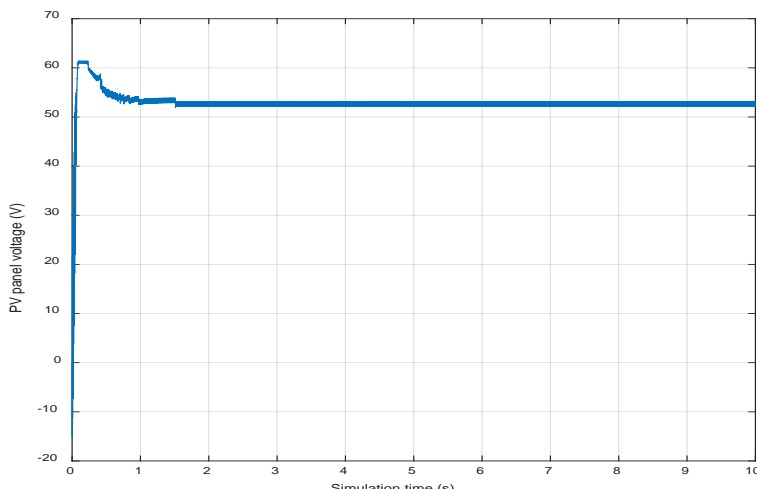

**Figure 6.** PV array output voltage at constant irradiation.

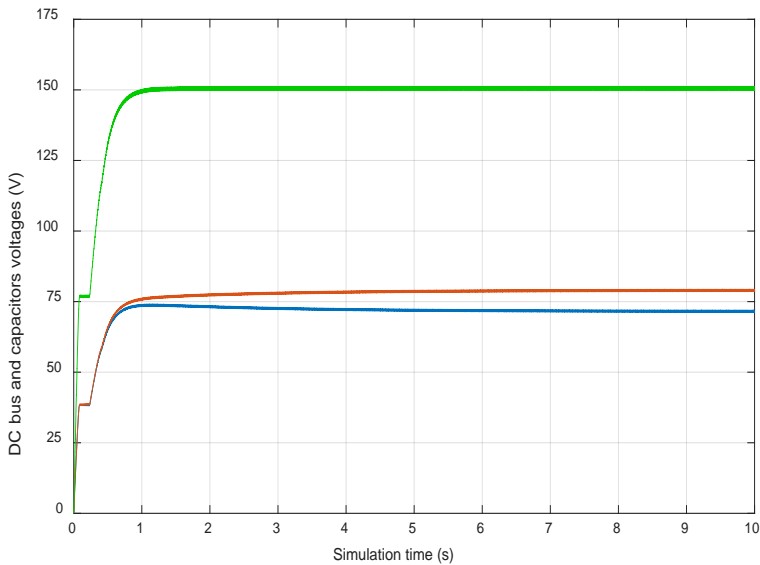

**Figure 7.** DC-link voltage evolution (the green curve) and capacitors voltages evolution (the red and blue curves).

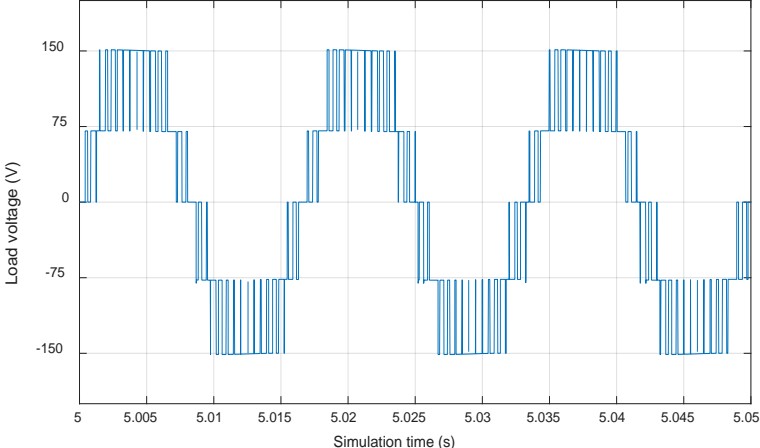

**Figure 8.** Waveform of the load voltage.

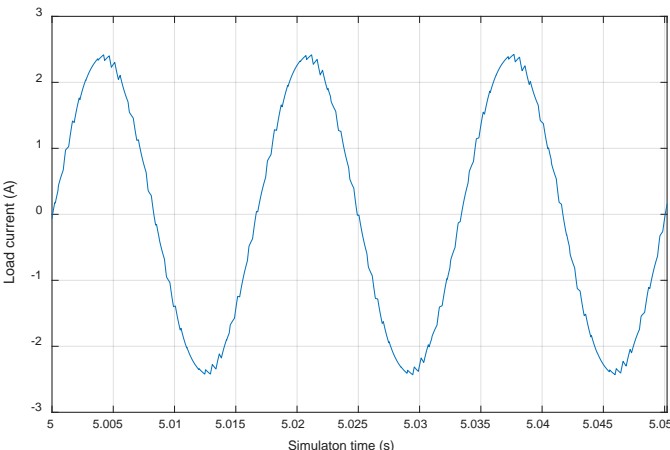

**Figure 9.** Waveform of the load current.

The proposed system is simulated under a constant temperature value of 25 °C and different levels of irradiation, as shown in Figure 10, to insure the efficiency and accuracy of the used INC technique. The simulation began with 750 W/m² until t = 4 s, then continued with 1000 W/m² of irradiation.

Figure 11 shows the evolution of the PV panel power. It is at the maximum values for both levels of irradiation (around 225 W for 750 W/m² and 305 W for 1000 W/m²). This is consistent with the characteristics curves of the PV module presented in Figure 4, which shows the accuracy and efficiency of the used INC algorithm.

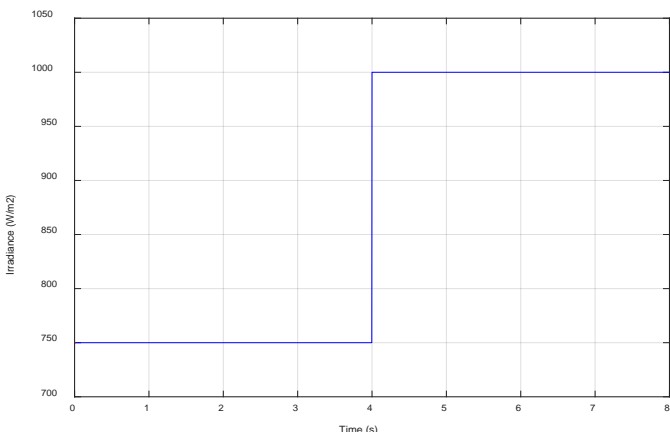

**Figure 10.** The variation of irradiance.

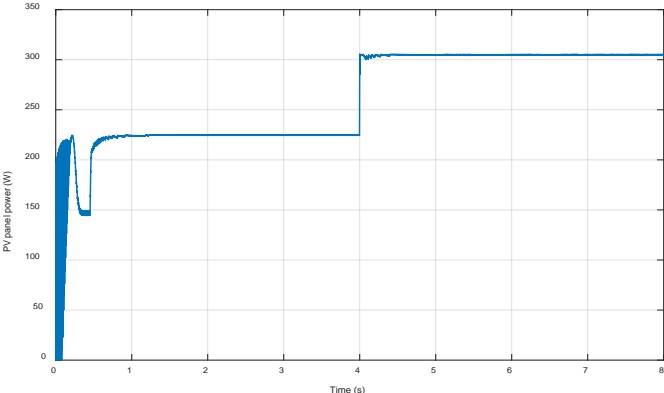

**Figure 11.** Waveform of the PV panel power under different levels of irradiation.

The same PV system is constructed and simulated using the PWM method used in [18] to control the SPUC5 inverter, in order to compare the efficiency of the proposed control method with this one by comparing the current THD generated by each PV system. Figure 12 presents the THD of the output current using the proposed control technique and it is around 1.96%, while Figure 13 shows the THD of the output current using the method presented in [18] and it is about 4.03%, which ensures the performance of the proposed control technique on the SPUC5 inverter.

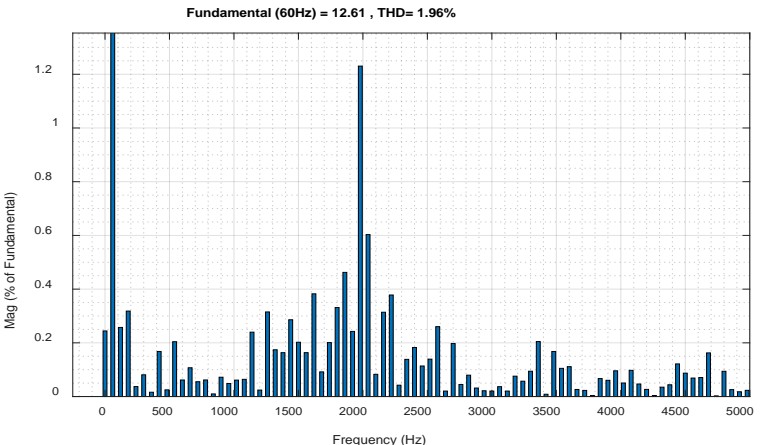

**Figure 12.** Load current THD spectrum using the proposed method.

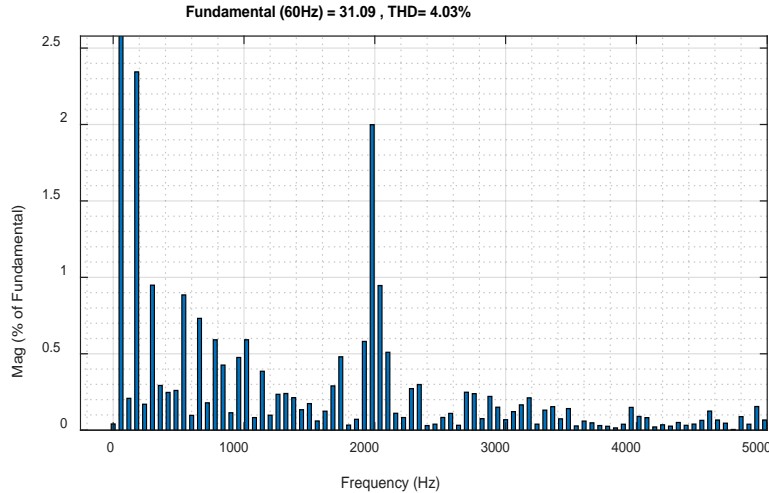

**Figure 13.** Load current THD spectrum using the traditional PWM method.

## 5. Experimental Validation

Experimental validation had been performed using the Dspace DS1103. Since no sensor or feedback is required for the SPUC5 inverter, hence, only five digital inputs/outputs are used to move the pulses out of the MOSFET gate. The MPPT technique, however, required an additional MOSFET and two sensors to acquire the PV array current and voltage. The experimental setup is depicted in Figure 14.

Figure 15 shows capacitances voltages which are well equilibrated about their reference, which is the half of the DC link voltage. These values are nearly the same as in Figure 7. The load current is nearly sinusoidal, whereas its values coincide with those in Figure 9. This offers the possibility to avoid using harmonics filters, which reduces the product cost. The output voltage of the inverter is also depicted in the same figure. The PV current and voltage evolution is shown in Figure 16. These values are in concordance with those of simulation, which validates the proposed concept experimentally.

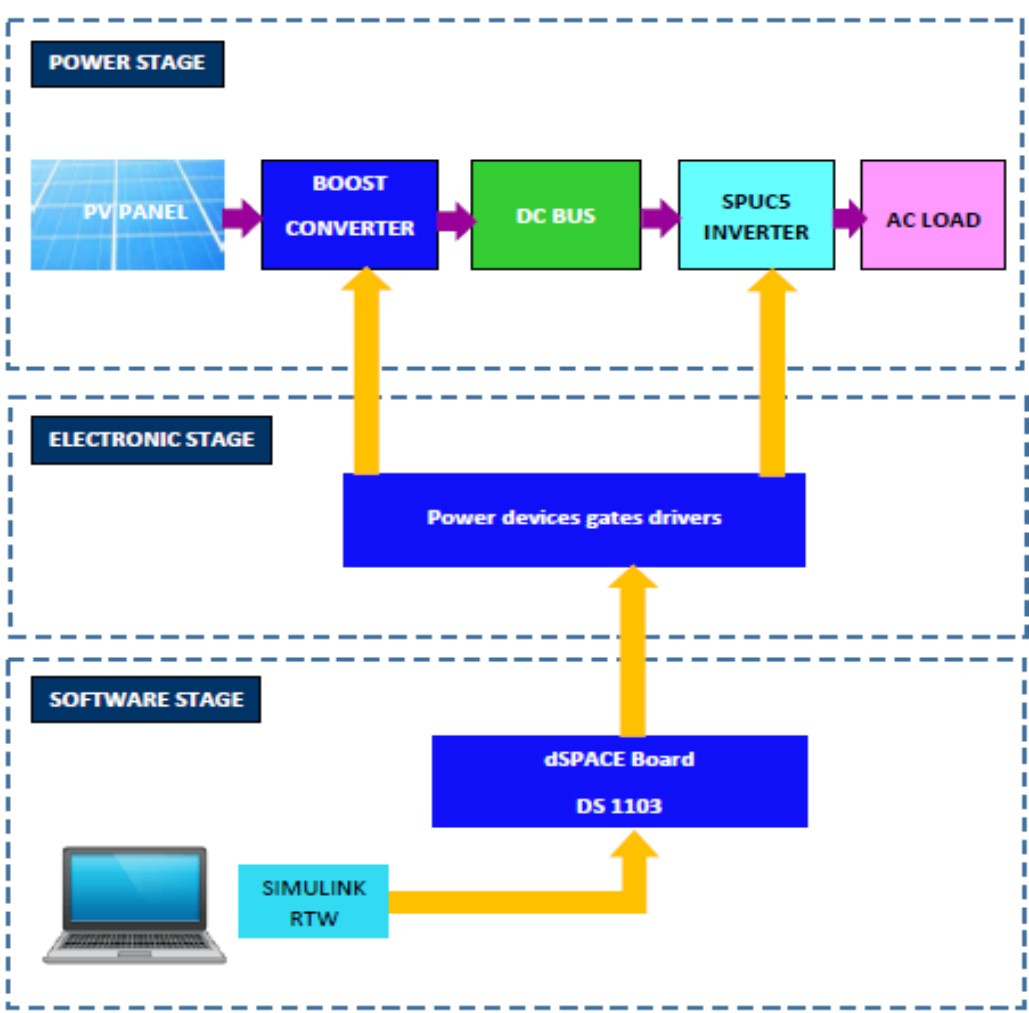

**Figure 14.** Experimental setup.

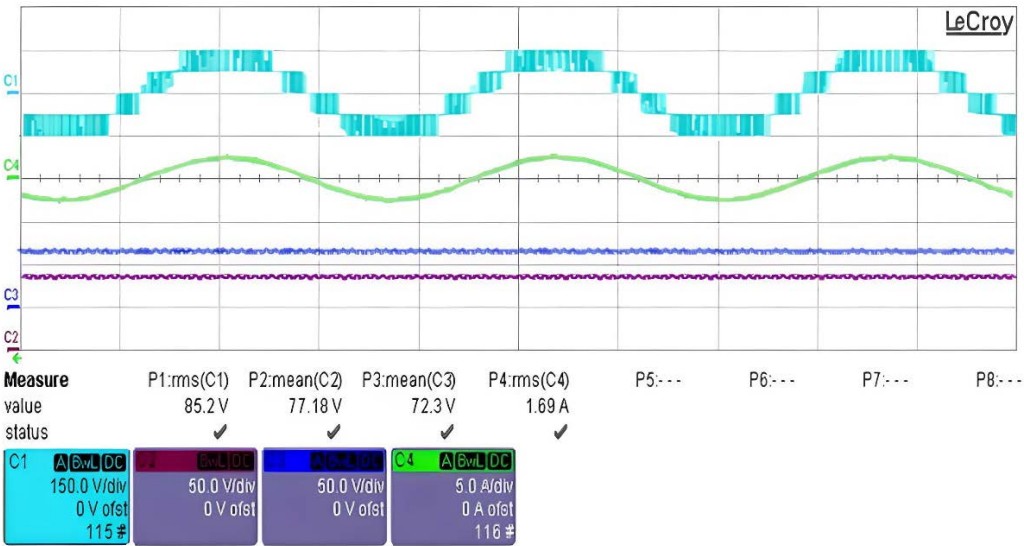

**Figure 15.** Experimental waveforms showing: output voltage of the inverter (C1), load current (C4) and capacitors voltages (C2 and C3).

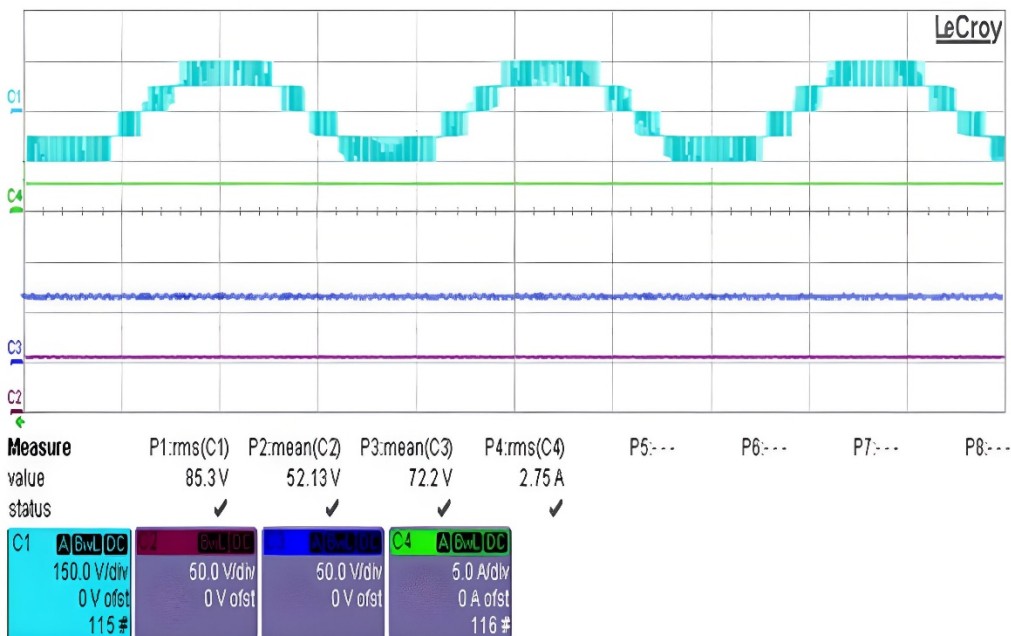

**Figure 16.** Experimental waveforms showing: inverter output voltage (C1), PV array voltage (C2), capacitor voltage (C3) and PV array current (C4).

## 6. Conclusions

A novel standalone PV system is presented in this paper. The incremental conductance has been preferred as the MPPT technique due to its high dynamics and rapidity in achieving the MPP, whereas, the SPUC5 has been proposed because it offers a high energetic efficiency conversion and it is well optimized in terms of component count. Moreover, the proposed PWM control technique allows for capacitor voltage balancing, without using any sensor or feedback, and ensures less THD rite compared to the traditional PWM method.

Simulation results have proven the high performance and the accuracy of the proposed system. Indeed, the tracking of the MPP was fast and very precise. Similarly, on the load side, the current waveform was almost sinusoidal due to the proposed PWM technique and the SPUC5 inverter. Therefore, there is no need to add filters, which reduces the product cost.

The experimental results ensure and validate the effectiveness of the advanced PWM used to control the SPUC5 inverter and all components and controls of the proposed PV system.

**Author Contributions:** Data curation, H.E.O.; Methodology, H.E.O.; Resources, H.E.O. and Y.O.; Software, H.E.O., A.E.G., Y.O. and K.A.-H.; Writing—original draft, H.E.O.; Writing—review and editing, Y.O. All authors have read and agreed to the published version of the manuscript.

**Funding:** This research was funded by l'école de technology supérieure, Montréal, Quebec, Canada.

**Conflicts of Interest:** The authors declare no conflict of interest.

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
