# Peer review of "Conception and Experimental Validation of a Standalone Photovoltaic System Using the SUPC5 Multilevel Inverter"

_electronics, doi:10.3390/electronics11172779_

Round 1
Reviewer 1 Report (Previous Reviewer 2)
I think the article is suitable for Electronics journal however some concern are existed for this reviewer.
There are so many variables in the manuscript, and a nomenclature is recommended to introduce all the variables and used in the paper.
Literature review on control of DC/AC converter for grid connected photovoltaic systems should be improved. Thus, the authors are invited to update the introduction and refer the following reference in the literature review: (i) Nonlinear Voltage Control for Three-Phase DC-AC Converters in Hybrid Systems: An Application of the PI-PBC Method, Electronics 9 (5), 847; Monitoring PWM signals in stand-alone photovoltaic systemsMeasurement 134, 412–425; (iii) LQR-based adaptive virtual inertia for grid integration of wind energy conversion system based on synchronverter model Electronics10(9), 1022
More result should be included to highlight the accuracy of the transient behavior.
Author Response
Please see the attachment.

Reviewer 2 Report (New Reviewer)
Comments to Authors:
1. The literature review of the paper is very limited. The authors must revise the introduction part and cite the latest work on the topic.
2. The conventional control techniques of the SPUC5 inverter are not discussed.
3. Fig. 2 must be labeled for its different elements.
4. The experimental waveforms are not clear. Authors must increase the image resolution of the obtained waveforms.
5. Comparative performance analysis of the conventional and proposed control techniques must be provided or discussed in the results.
Round 2
Reviewer 1 Report (Previous Reviewer 2)
I consider that my questions were addressed appropriately. I have no further comments. The paper can be published as it is.
Author Response
Comments and Suggestions for Authors : I consider that my questions were addressed appropriately. I have no further comments. The paper can be published as it is.
Response : thank you.
Reviewer 2 Report (New Reviewer)
The authors have addressed all the concerns raised by the reviewer and the paper is now accepted for publication.
Author Response
Comments and Suggestions for Authors : The authors have addressed all the concerns raised by the reviewer and the paper is now accepted for publication. Response : thank you.This manuscript is a resubmission of an earlier submission. The following is a list of the peer review reports and author responses from that submission.
Round 1
Reviewer 1 Report
the main contribution of this paper is not clear, is the contribution in control or in power stage?
SUPC is already published, what is the difference between this one and published topologies.
proposed control is MPPT or different control, there is no experimental results for MPPT only open loop output of the converter
Reviewer 2 Report
The robustness of the DC/AC converters is important for its functioning. Grid connected photovoltaic systems are important in the mix of energy sources. I think the article is suitable for Electronics journal however some concern are existed for this reviewer.
The Abstract in its current from is an alternative Introduction, it should clearly describe the scope with more focusing on the proposed approach and results of the study. Thus, the structure of the abstract need to be changed covering, overview, problem, methods, results, conclusion.
There are so many variables in the manuscript, and a nomenclature is recommended to introduce all the variables and used in the paper.
Introduction section should provide a critical analysis of the available and appropriate literature to identify an objective whose accomplishment will provide a significant contribution to the field. The research gap in the literature should be clearly exposed.
Several controllers are presented in literatures. What is the main contribution of your proposed method over the other existing techniques?
Literature review on control of DC/AC converter for grid connected photovoltaic systems should be improved. Thus, the authors are invited to update the introduction and refer the following reference in the literature review: (i) Nonlinear Voltage Control for Three-Phase DC-AC Converters in Hybrid Systems: An Application of the PI-PBC Method, Electronics 9 (5), 847; Monitoring PWM signals in stand-alone photovoltaic systems Measurement 134, 412–425; (iii) LQR-based adaptive virtual inertia for grid integration of wind energy conversion system based on synchronverter model Electronics10(9), 1022
Please could the authors shorten the section 3, do not include widely known content in the literature. Also, avoid including results in this section
More result should be included to highlight the accuracy of the transient behavior.
Reviewer 3 Report
The authors wrote in the abstract that they presented a new concept of single-phase inverter, proposed a novel control method, and carried out simulation and experimental studies.
The reviewer has several comments and questions.
- It is not clear from the text what the authors' own contribution and their individual development are.
- What is the new solution of the inverter topology that the authors write about in the abstract, and similarly what is new in the control concept? The authors gave schematics and no comments or description of the operation.
- equations 1-10, they did not explain what the symbols vr, Z1-Z4 mean.
- line 117 Why was SPR-305-WHT chosen and not another PV module?
- Does Figure 5 show simulation results or PV module catalogue data?
- Table 3. Why was the switching frequency set at 2kHz? The voltage parameters at which the simulation and experiment were performed are not given.
- completely incomprehensible for the reviewer is the use of static load RL, after all, the system is to work with the mains, this completely disqualifies these results!!! The current is sinusoidal because the load uses inductance. What will happen if we connect an inverter with such a voltage shape to the power line? What will the current look like? The line inductance is small, after all!
- What is the purpose of Figure 5, after all it says that a constant irradiance is used in the simulation
- Why is DC at 150 V?
- The simulation results are insufficient, many more cases should be considered, even if only a few different radiation levels!!!
- The simulations do not include a DC/DC converter and yet it is there in the experimental schematic.
- Figures 10 and 11 are illegible, please clearly mark the appropriate levels of voltages and currents.
- No comparison and analysis of the simulation and experimental results.
- No underlining of the author's elaboration of presented solution!
- In reviewer opinion the paper contains a lot of mistakes regarding the presented system, simulation results and experiments, therefore, I recommend rejecting it.